# Advancements in Finite Element Modeling for Cardiac Device Leads and 3D Heart Models

**DOI:** 10.3390/bioengineering11060564

**Published:** 2024-06-03

**Authors:** Anmar Salih, Farah Hamandi, Tarun Goswami

**Affiliations:** 1Department of Biomedical, Industrial and Human Factors Engineering, Wright State University, Dayton, OH 45435, USA; hamandi.3@wright.edu; 2Department of Orthopedic Surgery, Sports Medicine and Rehabilitation, Miami Valley Hospital, Dayton, OH 45409, USA

**Keywords:** CIED leads insulation, in vivo environment, finite element modeling, cardiac device leads, computational simulations, silicone insulation, polyurethane insulation, pacemaker leads, ICD leads, CRT leads, lead behavior

## Abstract

The human heart’s remarkable vitality necessitates a deep understanding of its mechanics, particularly concerning cardiac device leads. This paper presents advancements in finite element modeling for cardiac leads and 3D heart models, leveraging computational simulations to assess lead behavior over time. Through detailed modeling and meshing techniques, we accurately captured the complex interactions between leads and heart tissue. Material properties were assigned based on ASTM (American Society for Testing and Materials) standards and in vivo exposure data, ensuring realistic simulations. Our results demonstrate close agreement between experimental and simulated data for silicone insulation in pacemaker leads, with a mean force tolerance of 19.6 N ± 3.6 N, an ultimate tensile strength (UTS) of 6.3 MPa ± 1.15 MPa, and a percentage elongation of 125% ± 18.8%, highlighting the effectiveness of simulation in predicting lead performance. Similarly, for polyurethane insulation in ICD leads, we found a mean force of 65.87 N ± 7.1 N, a UTS of 10.7 MPa ± 1.15 MPa, and a percentage elongation of 259.3% ± 21.4%. Additionally, for polyurethane insulation in CRT leads, we observed a mean force of 53.3 N ± 2.06 N, a UTS of 22.11 MPa ± 0.85 MPa, and a percentage elongation of 251.6% ± 13.2%. Correlation analysis revealed strong relationships between mechanical properties, further validating the simulation models. Classification models constructed using both experimental and simulated data exhibited high discriminative ability, underscoring the reliability of simulation in analyzing lead behavior. These findings contribute to the ongoing efforts to improve cardiac device lead design and optimize patient outcomes.

## 1. Introduction

The human heart exhibits remarkable vitality, beating approximately 100,000 times each day, totaling 30 million beats annually, and a staggering 2.5 billion times over the course of an average lifetime [1]. Despite being just slightly larger than a clenched fist, it demonstrates incredible efficiency by pumping around 7000 L of blood daily, 2.5 million liters yearly, and 200 million liters throughout an individual’s lifespan [2]. An anatomical model of the human heart, constructed from computer tomography and magnetic resonance images, is depicted in Figure 1 [3,4].

The synchronized opening and closing of these valves, orchestrated by a combination of electrical and mechanical forces, ensures proper chamber filling and ejection. Any disruption, such as valve malfunction or irregular electrical signals, can lead to serious conditions like arrhythmias or heart failure [5]. Understanding the intricate relationship between electrical impulses and muscular contractions offers valuable insights into these complex cardiac phenomena, with the potential to revolutionize treatment for the millions afflicted by heart disease [6].

The primary techniques and avenues for modeling cardiac electrophysiology (EP) involve employing mathematical models to replicate the EP dynamics of the myocardium [7]. During the 1970s and 1980s, EP models predominantly relied on rule-based cellular automata, gradually transitioning to equation-based models encompassing cellular-level and tissue-level equations. At the cellular level, these equations adhere to the Hodgkin and Huxley formalism, characterizing cellular action potential and ionic currents through a system of nonlinear first-order ionic ordinary differential equations. Such equations delineate the kinetics of individual channels, pumps, exchangers, and their electrical interplays. Moreover, there is a growing adoption of Markov-type models to construct more biophysically grounded representations of ion channels. Data from patch-clamp experiments have facilitated the formulation of comprehensive mathematical frameworks for ionic currents, facilitating the development of highly precise EP models. These models are instrumental in simulating electrical activation patterns and conduction velocities of waves within the ventricular myocardium. They can be broadly estimated from electrocardiograms (ECG) or body surface potential maps (BSPM), or locally assessed via electrical activation maps (EAMs). Additionally, extracellular ion concentrations can be approximated and integrated into models based on measurements of blood electrolyte concentrations, albeit recognizing their significant temporal variability.

As leads used in cardiac implantable electronic devices age, the risk of malfunction increases, with reported malfunction rates reaching 40% after 8 years. This underscores the importance of assessing damage in cardiac devices, given the significant number of patients with cardiovascular diseases and implantable devices. The complexity of these devices presents challenges in damage assessment, but such analyses can provide valuable insights for improving future designs. Salih et al. [8] examined various damage features of leads such as surface deformation, burnishing, pitting, scratching, discoloration, delamination, insulation defects, coil damage, and abrasion. The study highlights the importance of understanding common damage patterns in cardiac leads to improve future device designs. Salih and Goswami [9] investigated the residual properties of silicone leads used in cardiac implantable electronic devices, comparing them with new leads. Results show significant decreases in load to failure, elongation to failure, ultimate tensile strength, and percentage elongation at 5 N after long-term in vivo exposure, while the modulus of elasticity increases over time. Recently, they examined leads using polyurethane and SI-polyimide as insulators, analyzing their mechanical properties after various durations of in vivo implantation [10]. Results indicate no significant degradation in load to failure, elongation, ultimate tensile strength, or modulus of elasticity over time, except for a significant decrease in the percentage of elongation at 5 N force after 60 months. After experimental work, computational simulation is crucial to complement and enhance the understanding gained. While experiments provide valuable real-world data, they are often limited by factors such as cost, time, and complexity. Computational simulations can fill these gaps by providing a detailed, virtual representation of the system being studied. In the investigation of lead behavior in cardiac devices, computational simulations can help predict long-term performance and assess the impact of different materials and designs, offering insights that may not be feasible through experimentation alone.

The use of the finite element method (FEM) and computational fluid dynamics (CFD) modeling in cardiovascular device design and development is well-established. These methods enable engineers to simulate the behavior of cardiovascular devices in a controlled virtual environment, providing insights into their performance and helping to optimize their design. However, previous studies have limitations that our work addresses, making it important in the field.

Morris et al. [11] highlighted the significance of computational fluid dynamics modeling in cardiovascular medicine, emphasizing its role in improving device safety and effectiveness. While their work advanced the understanding of cardiovascular device behavior, it may have been limited in addressing specific device design challenges. Tourek et al. [12] reviewed cardiac pacemaker lead designs for computational models in a VR environment, showcasing the impact of FEM modeling on pacemaker design optimization. However, their focus may have been more on the virtual reality aspect rather than on the comprehensive analysis of lead performance over time, which is a critical factor in device durability and patient safety.

Additionally, Morris et al. [13] demonstrated the utility of virtual fractional flow reserve from coronary angiography in modeling the significance of coronary lesions, indicating the potential of FEM and CFD modeling in optimizing coronary artery bypass graft surgery planning. Their work contributed to understanding the hemodynamic effects of coronary lesions, but it may not have fully explored the intricacies of device–tissue interactions that our study addresses. Furthermore, Gasser et al. [14] proposed a novel strategy to translate the biomechanical rupture risk of abdominal aortic aneurysms to their equivalent diameter risk, highlighting the role of FEM and CFD modeling in aneurysm treatment planning. While their approach was innovative, it might not have delved deeply into the detailed analysis of device performance and material behavior, a key aspect that needed to be investigated.

Simon et al. [15] conducted simulations of the three-dimensional hinge flow fields of a bi-leaflet mechanical heart valve (BMHV) under aortic conditions, focusing on the detrimental stresses on blood elements caused by these flows. Their study aimed to characterize these flows to identify underlying causes for thromboembolic complications associated with BMHVs. While their work contributed to improving heart valve design, particularly in understanding the flow patterns around the hinge, it may not have addressed the broader scope of device–organ interaction that our study encompasses. Jiménez and Davies [16] demonstrated the hemodynamically driven stent strut design, emphasizing FEM modeling’s role in stent design optimization. However, their focus may have been more on stent geometry rather than on the comprehensive analysis of material properties and long-term performance.

Chiu et al. [17] compared the thromboresistance of different ventricular assist devices (VADs), highlighting the impact of FEM and CFD modeling on blood pump design optimization. While their study contributed to improving blood pump design, particularly in understanding thrombogenic potential, it may not have fully explored the dynamic behavior of devices within the cardiovascular system. Chen et al. [18] analyzed the fluid–structure interaction in a model of an L-Type Mg alloy stent-stenosed coronary artery system, demonstrating the use of FEM analysis in stent design optimization. While their study advanced stent design, it may not have addressed the comprehensive analysis of the device–tissue interaction.

In this study, three-dimensional models of cardiac leads and the heart were created using SolidWorks and MIMICS 25.0, while ANSYS Workbench 2022R1 was utilized for simulations. The cardiac leads were inserted into the heart 3D model, and simulations were run to investigate changes in lead behavior over time, ranging from new leads to more than 100 months of in vivo exposure. This work provides insights into the long-term performance of cardiac leads in vivo, comparing our findings with previous experimental work to enhance the robustness of our results and highlight the evolution of lead behavior under in vivo conditions. This comparative analysis adds depth to our understanding of lead performance and underscores the importance of our work in advancing the field of cardiac device design and optimization.

The objectives of this research endeavor encompass the development and validation of finite element models of cardiac device leads, aiming to create accurate representations of their mechanical behavior under various loading and stretching conditions. Through computational simulations, we seek to investigate the long-term performance of these leads in vivo, simulating the effects of in vivo implantation and physiological conditions to assess their durability and reliability. A crucial aspect involves comparing simulated results with experimental data to validate the accuracy of our computational models, thus ensuring their fidelity in representing real-world scenarios. Furthermore, we aim to optimize the design of cardiac devices for improved performance, leveraging insights gained from computational simulations to identify design enhancements and material optimizations. By exploring the impact of material properties on the mechanical behavior of cardiac device leads, we aim to enhance our understanding of their interaction with heart tissue and contribute to the advancement of cardiac device technology, ultimately striving to develop safer, more reliable, and longer-lasting cardiac device leads.

## 2. Materials and Methods

### 2.1. Computational Simulations

In our study, we employed advanced software tools to develop detailed three-dimensional models of both cardiac leads and the heart. Using SolidWorks, we constructed intricate models of cardiac leads, including a silicon pacing lead, a Polyurethane ICD lead, and a Polyurethane CRT lead. These models were designed to accurately represent the geometry and material properties of the leads used in cardiac implantable electronic devices.

Simultaneously, we utilized MIMICS 25.0 to create a three-dimensional model of the heart, incorporating its various components such as the myocardium, valves, chambers, and vessels. This detailed heart model was essential for simulating the interaction between the leads and the heart tissue realistically.

For conducting simulations, we employed ANSYS Workbench 2022R1, a powerful finite element analysis software. The heart model was meshed as nonlinear and quadratic to capture its complex mechanical behavior accurately. This meshing approach allowed us to simulate large deformations and material nonlinearity characteristic of soft tissues like the heart, ensuring the fidelity of our simulations.

The cardiac leads were then inserted into the heart model, and simulations were run to investigate how the leads behaved over time. By analyzing changes in lead behavior, we gained valuable insights into the long-term performance of cardiac leads in vivo. This aspect of our study is crucial for understanding how leads interact with the heart tissue and how their mechanical properties may change over time, providing valuable information for improving the design and durability of cardiac implantable electronic devices.

### 2.2. Geometry Creation

Three-dimensional models of the cardiac leads: silicon pacing lead (d = 2 mm), Polyurethane ICD lead (d = 2.8 mm), and Polyurethane CRT lead (d = 1.75 mm) were constructed using SolidWorks (Dassault Systèmes SolidWorks Corp., Concord, MA, USA) and imported in ANSYS Workbench 2022R1 (ANSYS Inc., Canonsburg, PA, USA) to simulate the loading conditions and regions of stress development. The heart and its attached vessels were modeled as well. Firstly, imaging data of a healthy heart and its surrounding vessels were taken from DICOM Image Library [19]. Secondly, MIMICS 25.0 program was employed to convert the imaging data into a 3D model, following the same procedure used in our previous modeling [20], allowing for the differentiation between the various components of the heart, including the myocardium, mitral valve, tricuspid valve, aortic valve, pulmonary valve, left atrium, right atrium, left ventricle, right ventricle, ascending aorta, aortic arch, descending aorta, pulmonary artery, superior vena cava, inferior vena cava, and coronary arteries, as shown in Figure 2. This process involved segmenting the different structures based on their densities and anatomical features [21]. By segmenting the different structures based on their densities and anatomical features, we were able to create a detailed and realistic 3D model that accurately represented the complex geometry of the heart and its associated vasculature. This detailed model was essential for conducting accurate simulations and investigating the behavior of the cardiac leads in vivo. Finally, the 3D model was refined and optimized using SolidWorks. This step allowed for the incorporation of finer details and adjustments to ensure the model accurately represented the structure and mechanical properties of the heart and its attached vessels. The final model would therefore include both the external features of the heart and vessels as well as internal structures in the myocardium and the layers of the vessel walls. After constructing the models, the leads were inserted into the heart model, as shown in Figure 3. The lead is then inserted into the right ventricle and the tip of the lead was fixed near the septal area near the apex, as shown in Figure 4.

### 2.3. Finite Element Meshing

The mesh of the heart was configured as nonlinear and quadratic. This setup was chosen to accurately capture the complex behavior of the heart tissue under various loading conditions. Nonlinear elements allow for the simulation of large deformations and material nonlinearity, which are characteristic of soft tissues like the heart. The quadratic mesh type enhances the accuracy of the simulation by using higher-order interpolation functions for the elements, providing a more detailed representation of the geometry, and reducing numerical errors. This combination of nonlinear and quadratic meshing ensured that the simulation accurately captured the mechanical response of the heart tissue, enhancing the reliability and precision of the results. A mesh size of 0.5 mm was selected for both the heart and lead geometry as shown in Figure 5. This choice aimed to strike a balance between accuracy and computational efficiency. Each element in the mesh had a characteristic size of 0.5 mm, enabling the capture of complex geometrical features while maintaining manageable computational costs. A smaller mesh size would have provided more detailed geometry representation but at a higher computational expense. Conversely, a larger mesh size would have reduced computational costs but could have led to less accurate results, particularly in regions with intricate geometry or high stress gradients. By using the 0.5 mm mesh size, the simulation could accurately depict the behavior of the lead within the heart, ensuring meaningful and reliable results.

In this study, mesh convergence checks were performed to assess the sensitivity of the simulation results to changes in the mesh density. The number of elements and nodes in the mesh directly impacts the accuracy of the simulation, with finer meshes generally providing more accurate results but requiring more computational resources. The total number of elements and nodes for each model, including the heart with different leads, was carefully determined to ensure that the results were consistent within a five percent margin. This meant that further refinement of the mesh did not lead to significant changes in the results, indicating that the chosen mesh density was sufficient to capture the behavior of the system accurately. After performing mesh convergence checks, the total number of elements for the heart model with the pacing lead was 2,287,865, with 3,619,097 nodes. For the pacing lead itself, there were 4872 elements and 24,411 nodes. Correspondingly, the heart model alone consisted of 2,282,993 elements and 3,594,686 nodes. The total number of elements for the heart model with the ICD lead was 2,289,083, with 3,625,200 nodes. For the ICD lead itself, there were 6090 elements and 30,514 nodes. The total number of elements for the heart model with the CRT lead was 2,286,647, with 3,612,994 nodes. For the CRT lead itself, there were 3654 elements and 18,308 nodes, as shown in Table 1. By refining the mesh using a smaller increment size, the simulation was able to accurately depict the complex interactions between the leads and the heart tissue, providing valuable insights into the long-term performance of the cardiac leads in vivo. The meticulous meshing process ensured that the models were robust and capable of providing meaningful and reliable results for further analysis and interpretation. Table 2 shows the mesh convergence within a five percent margin for the polyurethane ICD lead inside the heart model with less than a month of in vivo exposure.

### 2.4. Material Assignment

The material properties of the new leads were assigned according to the ASTM standards D 1708-02a [22] and D 412-06a [23], which provide guidelines for determining the tensile properties of plastics and elastomers, respectively. These standards ensure that the material properties used in the simulation are based on well-established testing methodologies.

In contrast, the residual properties of the leads after in vivo exposure were derived from our previous experimental investigations [9,10]. These data were crucial for modeling the long-term behavior of the leads, as they reflects the actual performance of the leads after being implanted in patients for varying durations.

The in vivo implantation durations varied among the leads, ranging from new leads to 132 months. For pacing leads, the average in vivo duration was 55 ± 49 months, while for CRT leads, it ranged from less than a month to 108 months, with an average duration of 41 ± 31 months. ICD leads had in vivo durations ranging from less than a month to 89 months, with an average duration of 41 ± 27 months. These varying durations reflect the real-world conditions under which the leads are exposed to physiological stresses, and incorporating these data into the simulation ensures that the models accurately represent the behavior of the leads over time. The mechanical properties of cardiac muscle were crucial for accurately modeling its behavior under various mechanical loads and conditions in the ANSYS software 2022 R1. These properties were defined based on specific values: a Young’s elastic modulus of 80 kPa [24], which represents the stiffness and resistance to deformation of the muscle; an ultimate tensile strength of 110 kPa [24,25], indicating the maximum stress the muscle can withstand before failure; and a Poisson’s ratio of 0.4 [26], which shows the muscle’s tendency to contract laterally when stretched longitudinally. These properties were determined based on established values from the literature, ensuring that the model accurately represents the mechanical response of cardiac muscle. Incorporating these properties into the simulation allows for a more realistic representation of the behavior of the muscle under different loading conditions, providing valuable insights into its mechanical function and performance.

### 2.5. Loads and Boundary Conditions

In the simulation, the lead was fixed at its entry point into the heart to prevent unrealistic movement, representing its anchoring in reality. Other parts of the heart not in direct contact with the lead were also fixed to simulate the anchoring effect of surrounding tissues. To simulate the heartbeat, a displacement boundary condition was applied to the heart. This displacement was defined based on the expected motion of the heart during a cardiac cycle (Figure 6), incorporating the dynamics of the heart’s long axis. During ventricular ejection, the distance between the apex and base decreased rapidly, and the ventricles shortened by approximately 7 mm. This shortening plateaued towards the end systole to ensure enough blood was ejected. During ventricular filling, the long axis gradually returned to its initial length as the heart muscle relaxed.

The interaction between the lead and heart tissue was defined using a frictional contact formulation to handle potential nonlinear behavior. Contact properties, with a friction coefficient = 0.5, were specified to represent the contact between lead and heart tissue. A fine mesh, approximately 0.1 mm near the contact interface, was used to capture the contact behavior accurately. The contact interface was carefully aligned with the tissue surface to prevent unrealistic contact behavior. Solver settings were adjusted to ensure proper resolution of the contact interactions, with a convergence criterion and monitoring of contact forces and displacements. The contact model was validated by comparing simulation results with experimental data or the published literature to verify the accuracy of the contact forces and displacements predicted by the simulation.

Multiple simulations were then performed on these models to investigate the change in the residual properties with respect to the in vivo implantation period of the leads. This involved applying the defined loads and boundary conditions to simulate the interaction between the leads and the heart tissue over time. The simulations aimed to analyze how the mechanical properties of the leads changed over time due to the effects of in vivo implantation, providing insights into the long-term performance and durability of the leads in a realistic physiological environment.

## 3. Results

Data were collected and statistical analysis was performed. The experimental data were compared to simulation data to check for significance.

### 3.1. Silicone Insulation in Pacemaker Leads

A comprehensive assessment was conducted to compare the performance of the silicone insulator based on both experimental and simulated data, focusing on factors including its force-bearing capacity, ultimate tensile strength, and percentage elongation. Figure 7 shows the silicone insulation lead under FEM. The mean force tolerance of the insulator, derived from experimental results, was determined to be 19.6 N ± 3.6 N, with a maximum of 25.13 N, a minimum of 14.98 N, and a median of 19.5 N. Similarly, simulation data yielded a mean force of 19.6 N ± 3.3 N, with a maximum of 24.5 N, a minimum of 12.61 N, and a median of 19.5 N. Statistical analysis, depicted in Figure 8A, revealed no significant disparity between the two datasets (*p*-value = 0.99). The investigation also looked into the ultimate tensile strength of the insulator. Experimental findings indicated a mean ultimate tensile strength of 6.3 MPa ± 1.15 MPa, with a maximum of 8 MPa, a minimum of 4.77 MPa, and a median of 6.22 MPa. Similarly, simulation data exhibited a mean of 6.25 MPa ± 1.05 MPa, with a maximum of 7.8 MPa, a minimum of 4.01 MPa, and a median of 6.21 MPa. Notably, statistical analysis (*p*-value = 0.99), as illustrated in Figure 8B, demonstrated no significant differentiation between the experimental and simulated results. Furthermore, the examination encompassed the percentage elongation of the insulator. Experimental data revealed a mean percentage elongation of 125% ± 18.8%, with a maximum of 170%, a minimum of 99%, and a median of 125%. Similarly, simulation data displayed a mean percentage elongation of 124.9% ± 18.7%, with a maximum of 169%, a minimum of 98%, and a median of 124%. Statistical scrutiny (*p*-value = 0.91), illustrated in Figure 8C, affirmed the absence of noteworthy variance between the experimental and simulated data sets.

### 3.2. Polyurethane Insulation in ICD Leads

The polyurethane insulation covering the ICD lead underwent varied conditions, as shown in Figure 9. Analysis of the experimental data revealed a mean force of 65.87 N ± 7.1 N, with a maximum of 77.62 N, a minimum of 54.33 N, and a median of 66.7 N. In contrast, the simulated data exhibited a mean force of 69.74 N ± 5.1 N, with a maximum of 81.1 N, a minimum of 62.1 N, and a median of 70.45 N. Statistical evaluation, depicted in Figure 10A, even though there is a slight increase in the simulated values, indicated no significant difference in the simulated data (*p*-value = 0.18) compared to the experimental data. A further comparison was conducted between the experimental and simulated data concerning ultimate tensile strength (UTS). The mean experimental UTS stood at 10.7 MPa ± 1.15 MPa, with a maximum of 12.6 MPa, a minimum of 8.82 MPa, and a median of 10.8 MPa. Conversely, the mean UTS for the simulated data was notably higher at 11.32 MPa ± 0.8 MPa, with a maximum of 13.16 MPa, a minimum of 10.08 MPa, and a median of 11.42 MPa. Statistical analysis demonstrated no significance in the simulation data compared to the experimental results (*p*-value = 0.18), as depicted in Figure 10B. In terms of percentage elongation, the mean experimental value was determined to be 259.3% ± 21.4%, with a maximum of 290%, a minimum of 220%, and a median of 266%. Meanwhile, the mean percentage elongation for the simulated data was slightly lower, measuring at 242.7% ± 19.4%, with a maximum of 280%, a minimum of 212%, and a median of 247%. Statistical assessment revealed no significant difference between the two datasets (*p*-value = 0.08), as illustrated in Figure 10C.

### 3.3. Polyurethane Insulation in CRT Leads

Experimental and simulated tests were conducted to evaluate the polyurethane insulation of CRT leads, as shown in Figure 11. In the experimental assessment, the mean force measured was 53.3 N ± 2.06 N, with a range from 49.16 N to 56.39 N, and a median of 53.5 N. Conversely, in the simulated test, the mean force was slightly higher at 55.6 N ± 2.4 N, with values ranging from 52.4 N to 59.4 N, and a median of 55.07 N. Statistical analysis revealed a *p*-value of 0.0579, indicating no significant difference between the experimental and simulated data, although there was a marginal increase in the simulated results, as depicted in Figure 12A.

The ultimate tensile strength (UTS) was also investigated through both simulation and experimentation. The mean experimental UTS was determined to be 22.11 MPa ± 0.85 MPa, with a range from 20.4 MPa to 23.4 MPa, and a median of 22.2 MPa. Conversely, the mean simulated UTS was slightly higher at 23.07 MPa ± 1 MPa, with values ranging from 21.75 MPa to 24.65 MPa, and a median of 22.8 MPa. Statistical analysis yielded a *p*-value of 0.058, indicating no significant disparity between the experimental and simulated UTS data, as illustrated in Figure 12B.

Regarding percentage elongation, a crucial aspect of this study, the mean experimental percentage elongation was 251.6% ± 13.2%, with a range from 224% to 270%, and a median of 253%. Conversely, the mean simulated percentage elongation was slightly lower at 246.5% ± 10.8%, with values ranging from 226% to 260%, and a median of 251%. Statistical analysis resulted in a *p*-value of 0.4328, indicating no significant distinctions between the experimental and simulated data sets, as demonstrated in Figure 12C.

## 4. Discussion

This study presents advancements in finite element modeling for cardiac leads and 3D heart models, demonstrating close agreement between experimental and simulated data for silicone insulation in pacemaker leads, polyurethane insulation in ICD leads, and polyurethane insulation in CRT leads. This study highlights the effectiveness of simulation in predicting lead performance, with strong correlations between mechanical properties and high discriminative ability of classification models constructed using both experimental and simulated data. We conducted a comprehensive comparison between our simulation work and the previous experimental work conducted by Salih et al. [9,10]. This comparison aimed to assess the performance of insulation materials used in cardiac leads, specifically focusing on silicone insulation in pacing leads, polyurethane insulation in ICD leads, and polyurethane insulation in CRT leads. Salih et al.’s experimental work provided valuable insights into the mechanical properties and behavior of these insulation materials under various loading conditions and durations of in vivo exposure. By comparing our simulation results with their experimental data, we aimed to validate the accuracy and reliability of our simulation models in capturing the real-world behavior of these materials.

### 4.1. Silicone Insulation for Pacing Leads

The analysis of both experimental and simulated data reveals noticeable variability across different in vivo durations, suggesting dynamic changes in material properties or behavior over time. This variability is evident in the fluctuations observed in force, ultimate tensile strength (UTS), and percentage elongation throughout the study period. Such fluctuations indicate a complex interplay of factors influencing the material’s response under varying conditions. Despite these fluctuations, both experimental and simulated tests exhibit similar overall trends in force, UTS, and percentage elongation. However, there are discernible differences in the specific values obtained between the two sets of data. Notably, the simulated results consistently trend slightly lower than the experimental results across all measured parameters. This discrepancy suggests that while the simulation model captures essential aspects of the material behavior, there are inherent limitations or discrepancies in the model that lead to deviations from the experimental observations. These differences between experimental and simulated data can be attributed to several factors, including model assumptions, material properties, and boundary conditions. Figure 13 illustrates the influence of these factors on the observed disparities. For instance, variations in model assumptions may lead to inaccuracies in predicting material behavior, while differences in material properties used in the simulation compared to the actual material may affect the accuracy of the results. Additionally, discrepancies in boundary conditions between the simulated environment and the real-world experimental setup can further contribute to differences in observed outcomes. Despite these disparities, both experimental and simulated data exhibit variability across different in vivo durations. Specifically, both force and UTS demonstrate a decreasing trend over time, indicating potential degradation or weakening of the material. Percentage elongation also displays variability, with fluctuations observed across different durations. This dynamic behavior underscores the importance of considering temporal effects when analyzing material performance under varying conditions.

The aim was to investigate the relationship between three mechanical properties: force, ultimate tensile strength (UTS), and percentage elongation. Correlation analysis was conducted to determine whether these properties were related and to what extent. When it comes to the correlation between force and ultimate tensile strength (UTS), the analysis revealed a linear correlation between force and UTS. This implies that as force increases, there is a corresponding increase in UTS, and vice versa. This relationship is consistent with mechanical principles, as UTS represents the maximum force a material can withstand before breaking, and force directly impacts the material’s strength [27]. On the other hand, percentage elongation showed a positive correlation with both force and UTS. This suggests that materials with higher force and UTS tend to exhibit greater elongation before failure. This relationship highlights the material’s ability to deform plastically before reaching its breaking point, which is an important consideration in various engineering applications [28], as shown in Figure 14A.

After establishing correlations between mechanical properties, the performance of classifiers applied to both experimental and simulated data was assessed using the area under the curve (AUC) of the receiver operating characteristic (ROC) curve. Both the experimental and simulated data yielded an AUC of 0.6844, as shown in Figure 14B. This indicates that the classifiers applied to both types of data perform similarly in distinguishing between the two classes. The consistent performance of classifiers on both experimental and simulated data suggests that the simulation model effectively captures essential aspects of the material behavior, allowing for reliable predictions and classification [29].

### 4.2. Polyhurethane Insulation in ICD Leads

The mean force values obtained from both experimental and simulated data exhibit a close resemblance, with the simulated data slightly higher on average. Similarly, when comparing the mean UTS values between experimental and simulated data, they are found to be quite comparable, though with the simulated data showing a slight elevation. Moreover, the mean percentage elongation values also demonstrate similarity between the experimental and simulated datasets, albeit with marginally higher values observed in the experimental data. Despite these slight disparities between experimental and simulated results for force, UTS, and percentage elongation, the main trends and magnitudes remain largely consistent. This consistency suggests that the simulation effectively captures the behavior observed in experimental testing, even if with some minor deviations, as shown in Figure 15. These observed differences in values may stem from various factors such as model assumptions, discrepancies in material properties, and inherent measurement error. Model assumptions, in particular, play a crucial role in shaping simulated outcomes, and any inaccuracies or simplifications in these assumptions could lead to deviations from experimental results [30]. Furthermore, variations in material properties between the simulated and real-world materials may contribute to the observed differences. Additionally, the presence of measurement errors, minimized to the extent possible, could also introduce slight discrepancies between experimental and simulated data [31].

The data suggest strong positive correlations between force and UTS, as well as moderate positive correlations between percentage elongation and both force and UTS. These correlations provide insights into how changes in one variable relate to changes in another, which can be crucial for understanding the behavior of materials under different conditions, as shown in Figure 16A. This indicates a perfect positive correlation between force and UTS. In other words, as the force increases, UTS also increases proportionally, and vice versa. The correlation coefficient of 1.0000 suggests a strong linear relationship between these two variables. It implies that changes in force directly correspond to changes in UTS, following the same direction and magnitude. When it comes to the relationship between percentage elongation and UTS and force, the correlation coefficient of 0.6325 indicates a moderate positive correlation between percentage elongation and UTS. As UTS increases, percentage elongation tends to increase as well, though not as strongly as the correlation between force and UTS. The lower and upper bounds of the 95% confidence interval (0.2639 and 0.8399, respectively) suggest that this correlation is statistically significant.

The ROC curve was employed to evaluate the effectiveness of classification models constructed using two distinct datasets. One derived from experimental data and another from simulation data. The AUC values, reported as 0.9800 for both datasets, serve as indicators of how well these models can differentiate between different classes within their respective datasets. The high AUC values signify that the classification models exhibit exceptional performance in distinguishing between the classes present in each dataset. Specifically, a value of 0.9800 suggests that the models are highly adept at correctly identifying true positives while minimizing the occurrence of false positives. This indicates a high level of discrimination between the classes, reinforcing the reliability and effectiveness of the classification models. The consistency of the AUC values (0.9800) across both the experimental and simulation datasets further underscores the robustness and generalizability of the classification models. This consistency implies that regardless of whether the models are based on experimental or simulated data, they demonstrate equally high performance in distinguishing between the classes present in each dataset.

Force and UTS demonstrate a perfect positive correlation, indicating that as force increases, UTS also increases proportionally, and vice versa, as shown in Figure 16A. This strong correlation underscores the direct influence of force on the material’s ultimate tensile strength. There is a strong positive relationship between percentage elongation and UTS. As UTS increases, the material tends to elongate more before reaching its breaking point. However, this correlation is not perfect, implying the involvement of other factors influencing percentage elongation. Similarly, there is a strong positive relationship between percentage elongation and force. As the force applied to the material increases, it tends to elongate more before breaking. Like the previous correlation, this relationship is not perfect, suggesting the influence of other contributing factors. The narrow confidence intervals for both correlations between percentage elongation and UTS, and percentage elongation and force, indicate a high level of confidence in the estimated correlation coefficients, enhancing the reliability of the observed relationships.

### 4.3. Polyurethane Insulation in CRT Leads

When comparing the experimental and simulated results, we observe consistent patterns across all measured parameters, including force, ultimate tensile strength (UTS), and percentage elongation as shown in Figure 17. Both sets of data generally follow the same trend, showing an increase or decrease in values over the range of measurements. Despite some minor differences, such as the simulated values being slightly higher than the experimental values, the overall trends remain consistent between the two datasets. The disparities between experimental and simulated results can be attributed to model assumptions. Simulation models often make simplifications and assumptions about the behavior of materials and systems, which may not perfectly align with real-world conditions. Variations in material properties, such as elasticity or strength, can affect how a material responds to external forces in experimental versus simulated environments [32]. Differences in how boundary conditions are implemented in experiments versus simulations can lead to discrepancies in the observed outcomes [33]. Both experimental and simulated data may be subject to measurement error, introducing uncertainties in the results. Despite the differences between experimental and simulated results, the data suggest that simulation provides reasonably comparable results to experimental testing. It indicates the potential usefulness of simulation as a predictive tool for evaluating the performance of polyurethane insulation in CRT leads. By leveraging simulation, researchers and engineers can gain insights into the behavior of materials and systems under various conditions, enabling more informed decision-making in product development and optimization processes.

With an AUC of 0.8594 for both experimental and simulation data types, it indicates that both sets of data have a relatively high discriminative ability. This means that the models or methods used to generate the ROC curves, whether based on experimental data or simulation data, are effective at distinguishing between different categories or classes within the dataset. Essentially, the ROC curves generated for both the experimental and simulated data perform similarly well in their ability to differentiate between different classes or categories within the data. This suggests that both the experimental approach and the simulation approach are effective at capturing relevant patterns or characteristics within the dataset, allowing for meaningful discrimination between different groups or conditions. Therefore, the similarity in AUC values between the experimental and simulated data types indicates that both methodologies yield comparable results in terms of their discriminative power, further validating the usefulness and reliability of both approaches in analyzing and interpreting the data at hand. There is also a perfect positive correlation between UTS and force. This suggests that as UTS increases, the force also increases linearly. This positive correlation indicates that stronger materials tend to require more force to break. However, there is a weak negative correlation between UTS and percentage elongation (−0.1545). This suggests that as the UTS increases, percentage elongation tends to decrease slightly, although the correlation is not very strong. Force also has a perfect positive correlation with itself, as expected the negative correlation between force and percentage elongation (−0.1545) suggests that as force increases, percentage elongation tends to decrease slightly, as shown in Figure 18A.

### 4.4. Fatigue Cycle

The fatigue cycle of materials like polyurethane and silicone refers to their behavior under repeated loading and unloading over time. While these materials are known for their durability and flexibility, they can still experience fatigue when subjected to cyclic stresses [34]. The ultimate tensile strength applicable to insulation in pacing, ICD, and CRT leads can exhibit a considerable range, spanning from approximately 6 MPa to over 24 MPa [9,10]. Consequently, the evaluation of these insulation materials’ performance in vivo entails subjecting them to cyclic stress where each cycle corresponds to one heartbeat. This testing methodology allows for the assessment of how well the insulation withstands the repetitive stresses experienced during normal cardiac function. In engineering, understanding how materials and structures respond to varying loads is critical for predicting their performance and ensuring safety in real-world applications. Therefore, applying cyclic loading to test the performance of these insulation materials in changing environmental conditions is important, as shown in Figure 19.

As shown in Figure 20A, silicone is subjected to a cyclic stress representing 80% of its maximum capacity. Remarkably, even under this relatively high level of stress, silicone demonstrates exceptional durability, enduring a remarkable 2 × 10^7^ loading cycles before showing signs of fatigue or failure. However, as the cyclic stress level decreases to 80% of its original value, the endurance of silicone diminishes. At this reduced stress level, silicone can only withstand a significantly lower number of loading cycles, specifically 2.5 × 10^6^ cycles, before exhibiting signs of fatigue or failure. On the other hand, polyurethane showed a higher number of cycles and performance than silicone. Figure 20B shows the performance of polyurethane in CRT leads and involves understanding how the material responds to cyclic loading at different stress levels. In the beginning, polyurethane is subjected to a cyclic stress level that exceeds its nominal or expected loading capacity by 110%. Despite this slight overloading, the polyurethane demonstrates robust endurance, withstanding an impressive 2 × 10^7^ loading cycles. This suggests that polyurethane has a considerable margin of safety when subjected to slightly elevated stress levels. As the loading history increases to 150% of the nominal stress level, representing a more significant overload condition, the endurance of the polyurethane diminishes. Despite this increased stress level, the material still displays notable durability, withstanding 1.7 × 10^7^ loading cycles before showing signs of fatigue or failure. The data presented in Figure 20C indicate that the polyurethane insulation of ICD leads exhibits remarkable performance when subjected to cyclic loading. Initially, the polyurethane insulation withstands a loading history that exceeds its nominal stress level by 20%. Even under this heightened stress condition, the material demonstrates exceptional durability, enduring a substantial 2 × 10^7^ loading cycles before exhibiting signs of fatigue or failure. This emphasizes the material’s robustness and ability to withstand moderately elevated stress levels while maintaining structural integrity. When the loading history increases to 150% of the nominal stress level, representing a significant overload condition, the endurance of the polyurethane insulation decreases. Despite this more pronounced stress elevation, the material still showcases notable resilience, enduring 1.7 × 10^7^ loading cycles before reaching fatigue or failure.

### 4.5. Study Limitations

We acknowledge several limitations of our research. Firstly, our model made certain simplifications to enhance computational efficiency, such as neglecting certain anatomical details or simplifying material properties. While these simplifications were necessary for the scope of this study, they may limit the model’s ability to fully capture all aspects of cardiac behavior. Additionally, the generalizability of our findings may be limited, as our study focused on specific cardiac device leads and heart models. The findings may not be directly generalizable to other types of leads or patient populations. Finaly, due to the complexity of the heart and the limitations of current computational models, there is still much to explore in terms of improving the accuracy and applicability of such simulations. Future research could focus on refining the model with more detailed anatomical and physiological data, as well as incorporating patient-specific information for personalized simulations.

## 5. Conclusions

This study developed advanced finite element models for cardiac device leads and 3D heart models, which accurately capture the complex interactions between leads and heart tissue. This approach represents a significant innovation in the field of cardiac device design and evaluation. By utilizing ASTM standards and in vivo exposure data for material properties assignment, we have ensured that our simulations are realistic and applicable to real-world scenarios. Our work also demonstrates a strong correlation between mechanical properties, providing valuable insights for future research and design considerations. This research demonstrates the advancement in finite element modeling for cardiac device leads and 3D heart models. Through computational simulations using SolidWorks, MIMICS, and ANSYS Workbench, we created detailed models of cardiac leads and the heart to investigate their long-term performance in vivo. By meshing the heart as nonlinear and quadratic, we accurately captured its complex behavior under various loading conditions. Material properties were assigned based on ASTM standards and in vivo exposure data, allowing for realistic simulations of lead behavior over time. Our results indicate that the simulation models effectively capture the mechanical properties of cardiac leads, with close agreement between experimental and simulated data. Despite some minor differences, the overall trends remain consistent, suggesting that simulation provides valuable insights into lead performance and durability. Correlation analysis revealed strong relationships between force, ultimate tensile strength (UTS), and percentage elongation, highlighting the interplay between these mechanical properties. Classification models constructed using experimental and simulated data demonstrated high discriminative ability, indicating the reliability and effectiveness of both approaches in analyzing lead behavior. The ROC curves further validated the robustness of the models, emphasizing their utility in distinguishing between different classes within the datasets.

## Figures and Tables

**Figure 1 bioengineering-11-00564-f001:**
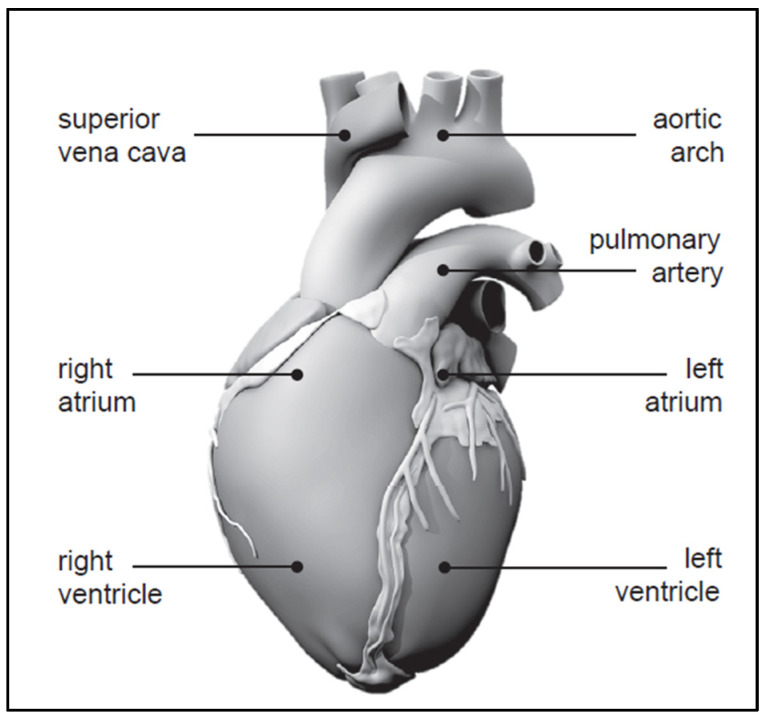
Human heart with four chambers [3,4].

**Figure 2 bioengineering-11-00564-f002:**
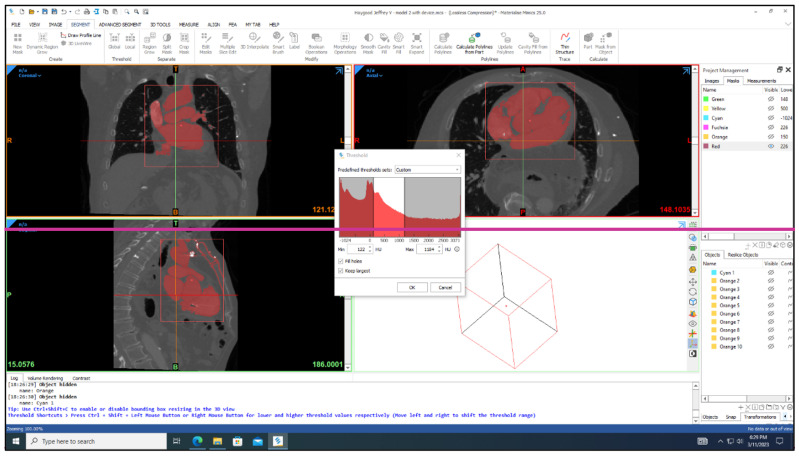
MIMICS 25.0 program was employed to convert the imaging data into a 3D model of the heart.

**Figure 3 bioengineering-11-00564-f003:**
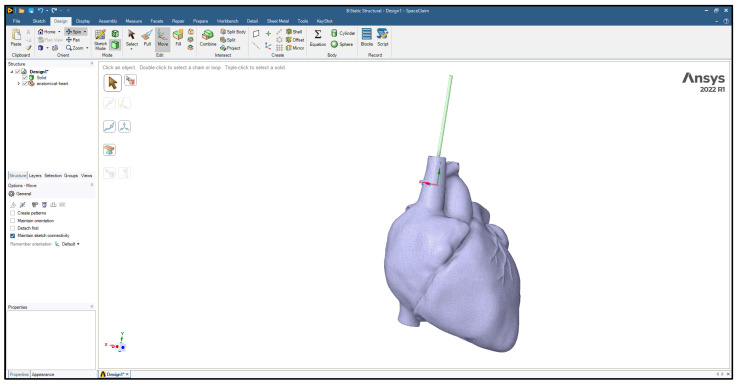
The lead inserted into the heart model.

**Figure 4 bioengineering-11-00564-f004:**
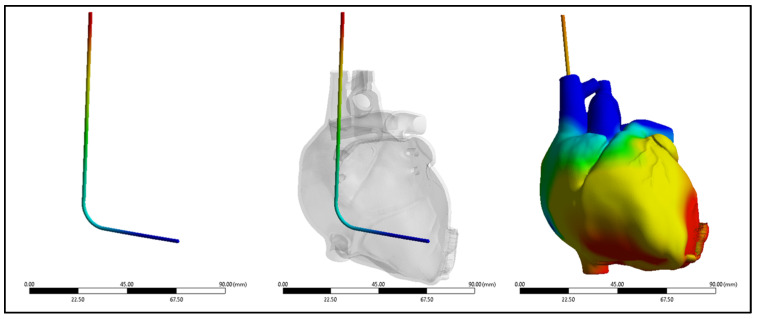
Lead insertion into the right ventricle.

**Figure 5 bioengineering-11-00564-f005:**
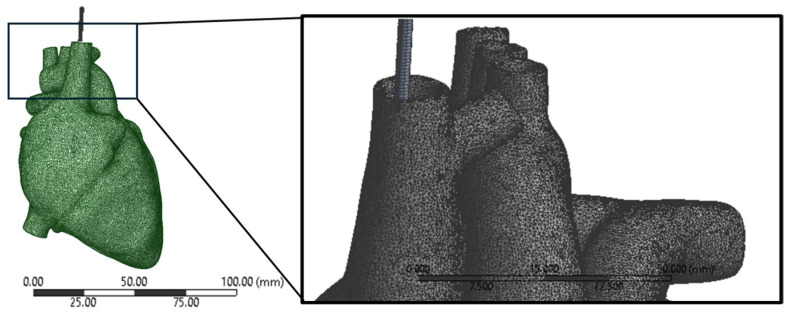
Meshing of the lead and heart models showing the quadratic mesh type.

**Figure 6 bioengineering-11-00564-f006:**
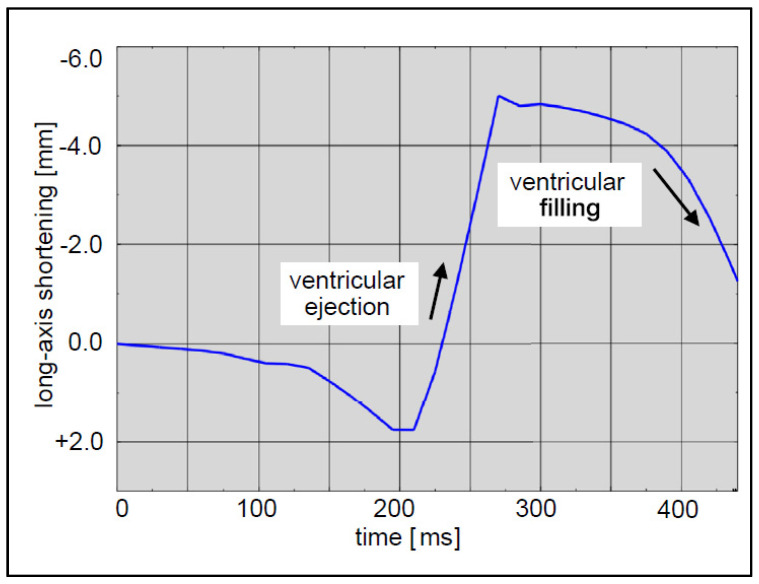
Expected motion of the heart during a cardiac cycle.

**Figure 7 bioengineering-11-00564-f007:**
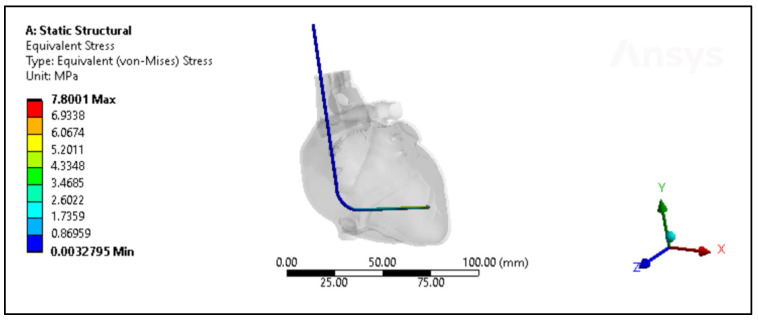
The von Mises stresses distribution of the new silicone pacing lead inside the heart.

**Figure 8 bioengineering-11-00564-f008:**
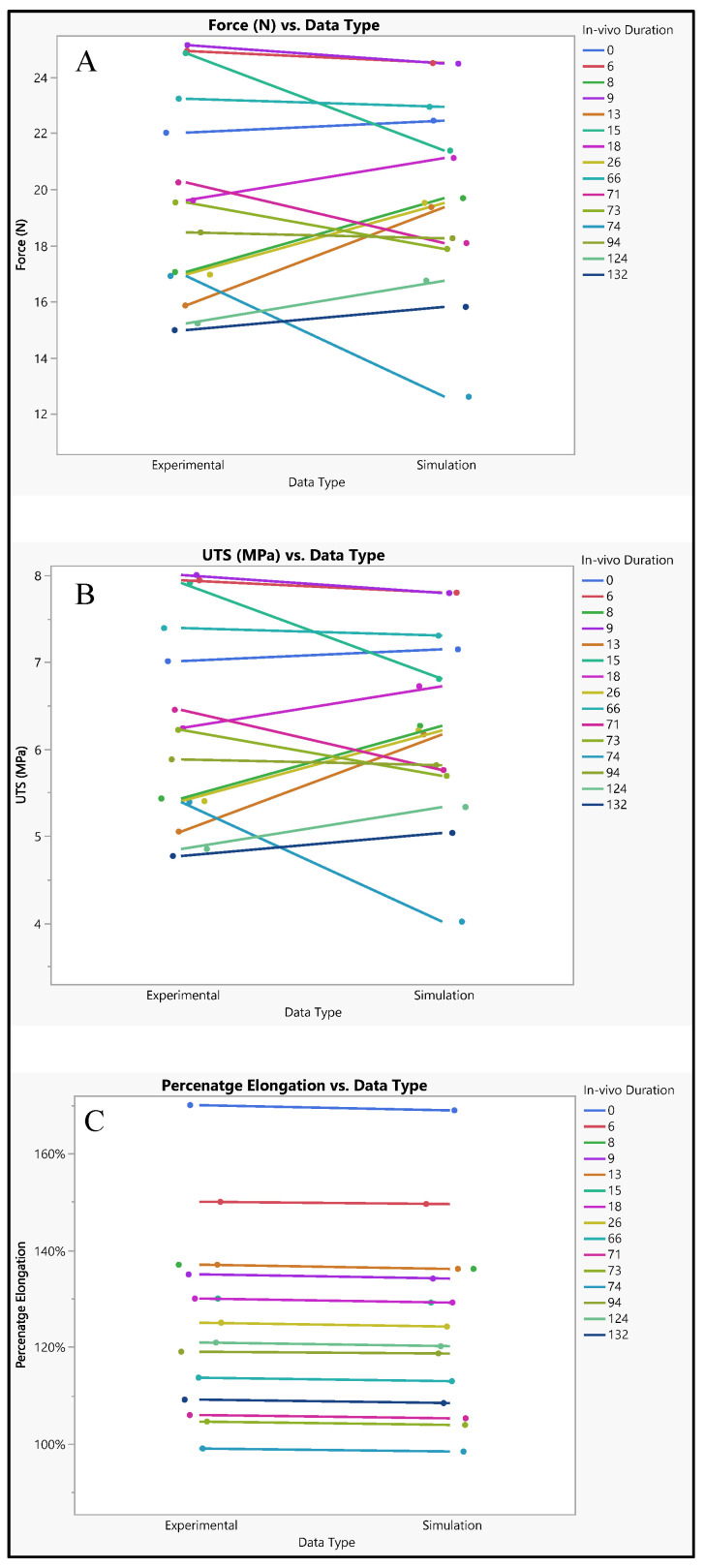
Experimental versus simulated data sets of silicone insulation for different parameters: (**A**) force in N, (**B**) ultimate tensile strength, and (**C**) percentage elongation.

**Figure 9 bioengineering-11-00564-f009:**
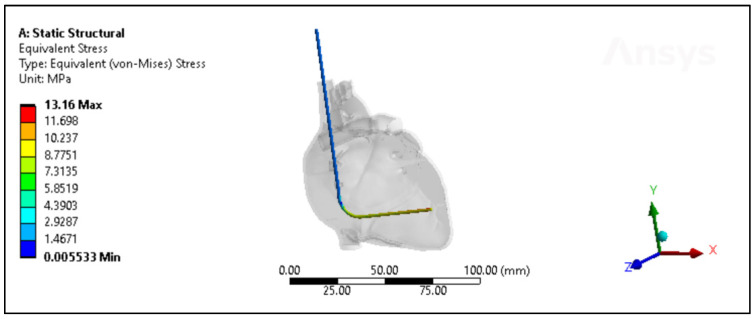
The von Mises stresses distribution of the Polyurethane ICD lead inside the heart with less than a month of in vivo exposure.

**Figure 10 bioengineering-11-00564-f010:**
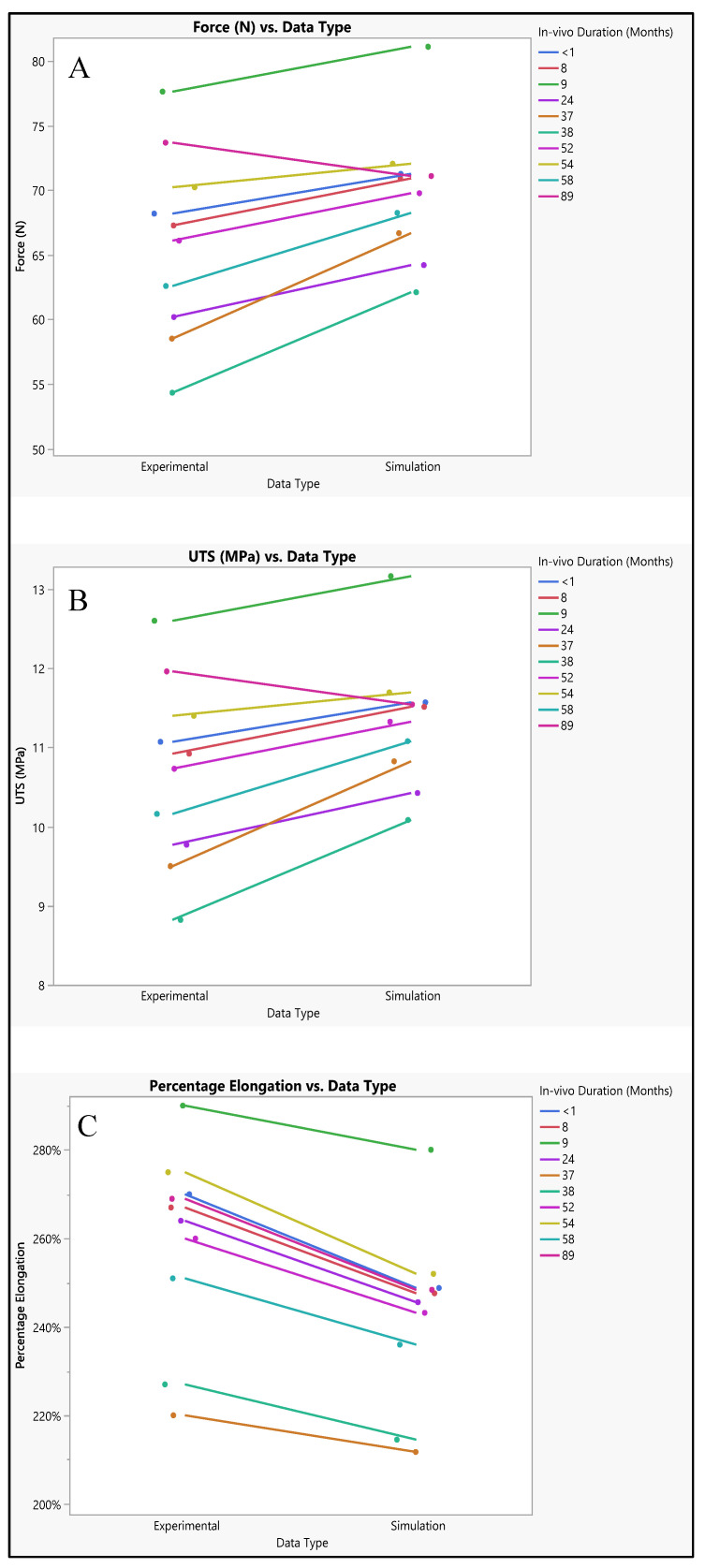
Experimental versus simulated data sets of polyurethane insulation for ICD leads for different parameters: (**A**) force in N, (**B**) ultimate tensile strength, and (**C**) percentage elongation.

**Figure 11 bioengineering-11-00564-f011:**
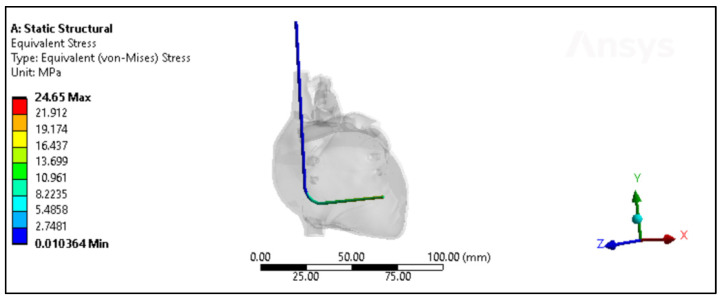
The von Mises stresses distribution of the Polyurethane CRT lead inside the heart model with less than a month of in vivo exposure.

**Figure 12 bioengineering-11-00564-f012:**
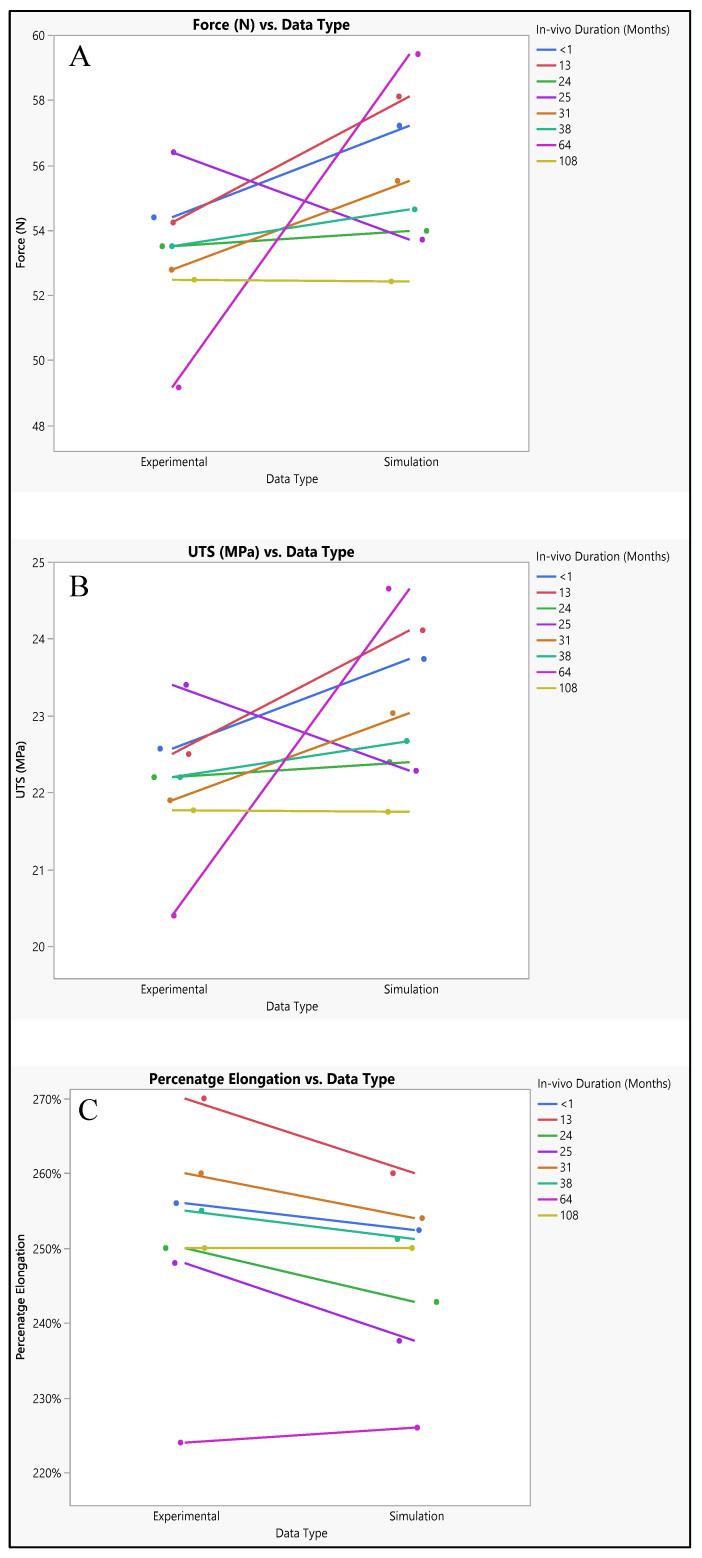
Experimental versus simulated data sets of polyurethane insulation for CRT leads for different parameters: (**A**) force in N, (**B**) ultimate tensile strength, and (**C**) percentage elongation.

**Figure 13 bioengineering-11-00564-f013:**
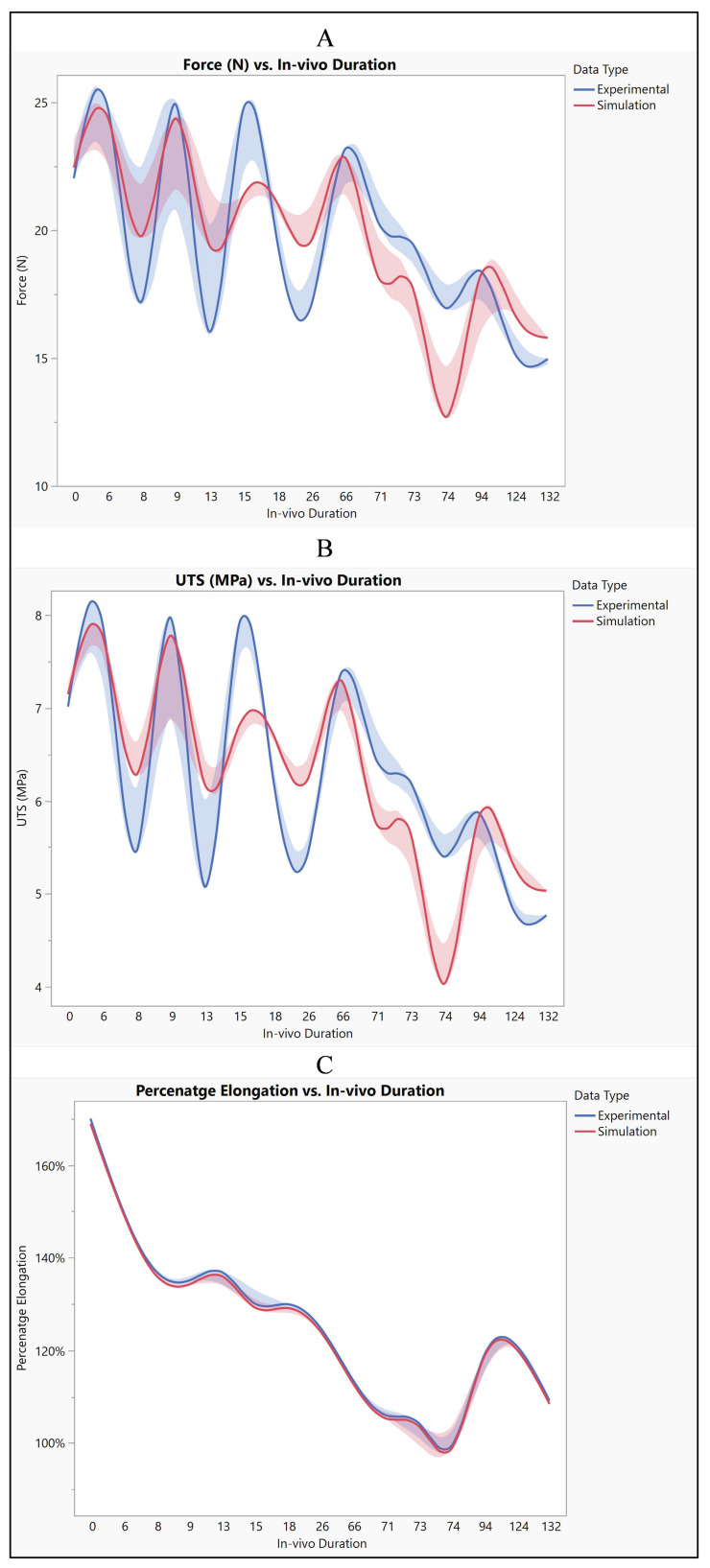
Silicone insulation experimental vs simulation (**A**) force, (**B**) UTS, and (**C**) elongation.

**Figure 14 bioengineering-11-00564-f014:**
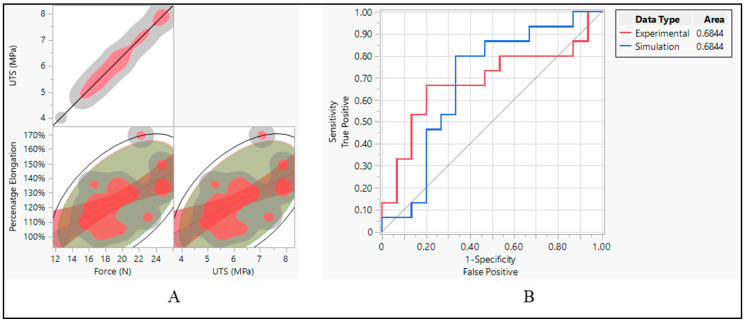
Silicone insulation pacing leads. (**A**) Correlation between force, UTS, and elongation and (**B**) ROC performance curve.

**Figure 15 bioengineering-11-00564-f015:**
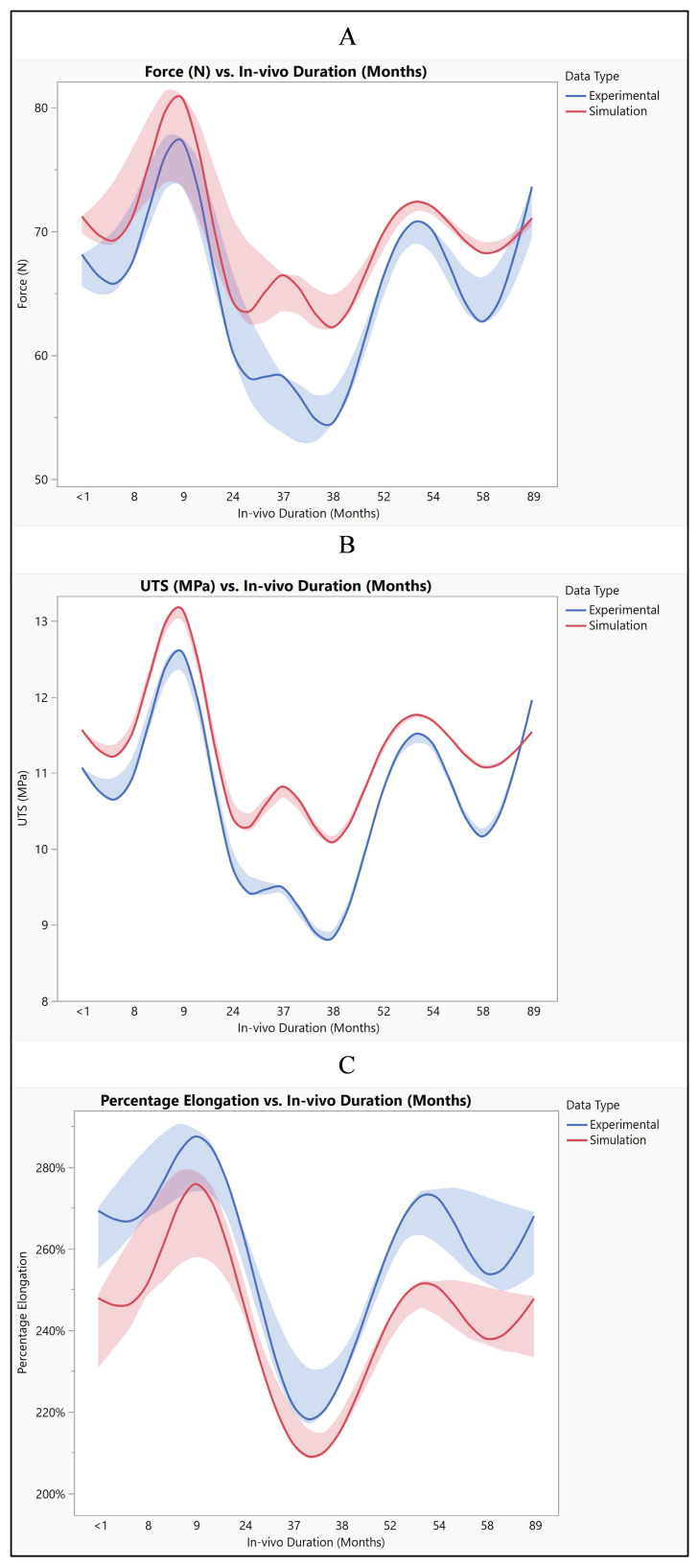
Polyurethane insulation for ICD leads experimental vs simulation: (**A**) force, (**B**) UTS, and (**C**) elongation.

**Figure 16 bioengineering-11-00564-f016:**
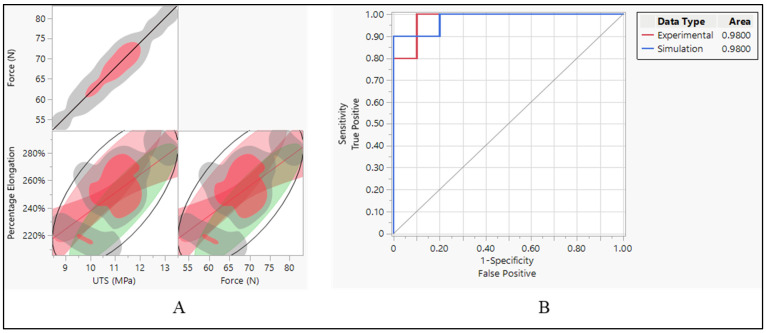
Polyurethane insulation ICD leads. (**A**) Correlation between force, UTS, and elongation and (**B**) ROC performance curve.

**Figure 17 bioengineering-11-00564-f017:**
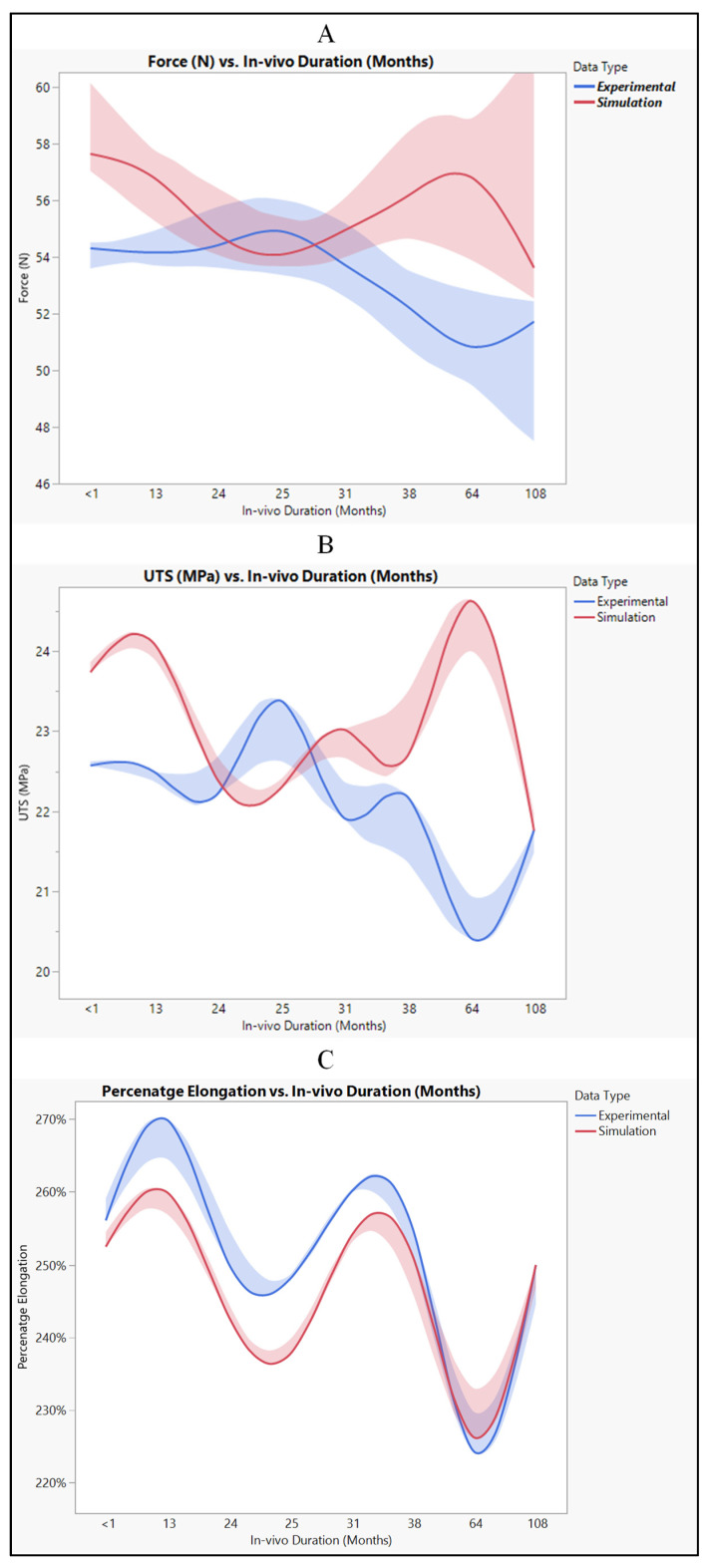
Polyurethane insulation for CRT leads, experimental vs simulation: (**A**) force, (**B**) UTS, and (**C**) elongation.

**Figure 18 bioengineering-11-00564-f018:**
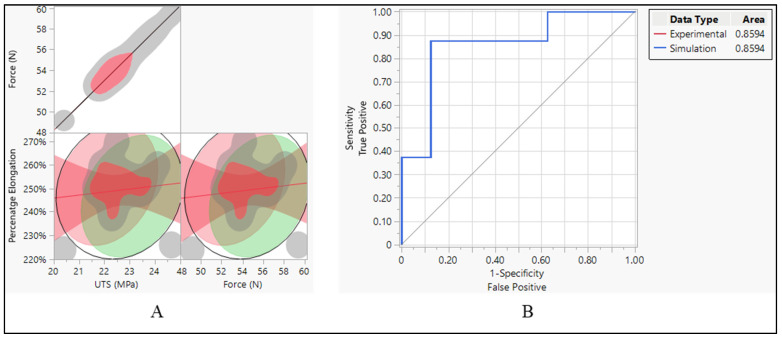
Polyurethane insulation CRT leads. (**A**) Correlation between force, UTS, and elongation and (**B**) ROC performance curve.

**Figure 19 bioengineering-11-00564-f019:**
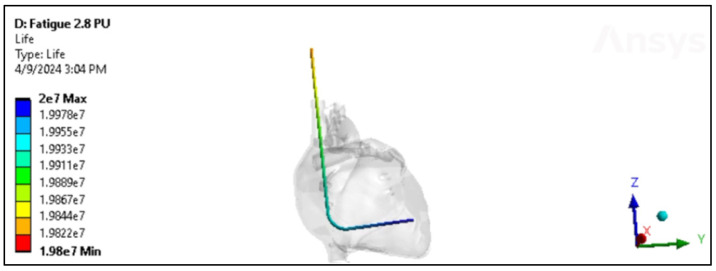
Fatigue in the Polyurethane ICD lead inside the heart.

**Figure 20 bioengineering-11-00564-f020:**
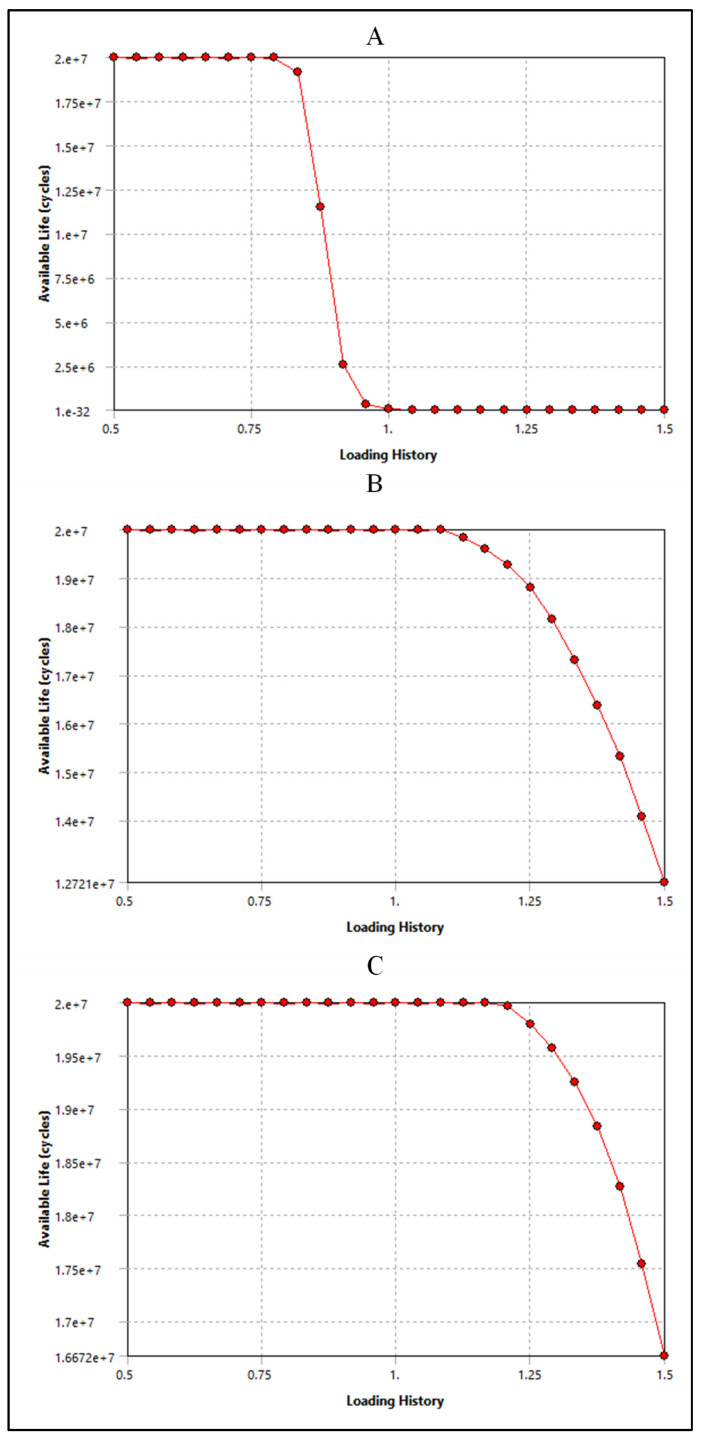
Sensitivity plot for (**A**) silicone insulation in pacemaker leads, (**B**) polyurethane insulation in CRT leads, and (**C**) polyurethane insulation in ICD leads.

**Table 1 bioengineering-11-00564-t001:** Number of elements and nodes in each model.

The Model	Elements	Nodes
3D model of the Heart	2,282,993	3,594,686
Pacing lead inside the heart	2,287,865	3,619,097
Pacing lead only	4872	24,411
ICD lead inside the heart	2,289,083	3,625,200
ICD lead only	6090	30,514
CRT lead inside the heart	2,286,647	3,612,994
CRT lead only	3654	18,308

**Table 2 bioengineering-11-00564-t002:** Mesh convergence within a five percent margin for a Polyurethane ICD lead inside the heart model with less than a month of in vivo exposure.

	Equivalent Stress (MPa)	Change (%)	Nodes	Elements
1	12.766		2,754,976	1,590,323
2	12.953	1.454	3,518,514	2,187,949
3	13.112	1.220	3,563,272	2,246,783
4	13.151	0.297	3,565,218	2,249,341
5	13.158	0.053	3,602,743	2,273,356
6	13.16	0.015	3,625,200	2,289,083

## Data Availability

The original contributions presented in the study are included in the article, further inquiries can be directed to the corresponding author/s.

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
