# Peer review of "Advancements in Finite Element Modeling for Cardiac Device Leads and 3D Heart Models"

_bioengineering, 2024, doi:10.3390/bioengineering11060564_

Round 1

Reviewer 1 Report

Comments and Suggestions for Authors

The authors investigate the load behavior of 3D heart models. The paper is well-written and interesting, and it can be considered for publication after the authors have replied to the following mandatory remarks:

  1. The authors should highlight the innovative contribution of their research activity.
  2. The authors should discuss the correct characterization of the boundary conditions in hemodynamic simulations, which have to correctly reproduce the effect of organs and vessels outside the simulated portion (see, e.g., Morbiducci et al., 2013; Mariotti et al., 2021). The authors should address the impact of inlet/outlet boundary conditions.
  3. The authors state in line 131 that a mesh convergence check has been performed. They should provide further detail, such as a table reporting that some quantities of interest do not change further by increasing the number of nodes.
  4. The authors could add a description of the velocity fields, also with the help of instantaneous indicators of the vortices, and a corresponding analysis of the characteristic frequencies.

Suggested references: 

Mariotti, A. et al. (2021) Hemodynamics and stresses in numerical simulations of the thoracic aorta: Stochastic sensitivity analysis to inlet flow-rate waveform. Comput. Fluids 230, 105123. 

Morbiducci, U. et al. (2013) Inflow boundary conditions for image-based computational hemodynamics: Impact of idealized versus measured velocity profiles in the human aorta. J. Biomech. 46(1), 102-109.

Author Response

Dear Reviewer,

Thank you for your insightful comments and positive feedback on our article. We appreciate your efforts in reviewing our work and are grateful for your constructive suggestions. Below, you will find our responses to each of your comments. We have addressed these comments both in the manuscript and in this response report in “red”.

Reviewer 2 Report

Comments and Suggestions for Authors

The finite element modeling was improved for cardiac device leads and 3D heart models. The authors provided many figures to facilitate the understanding of the research work. There are various key comments to be considered by the authors before the recommendation of paper acceptance.
Comment 1. Abstract:
(a) Define the acronym ASTM.
(b) Share numerical results.
Comment 2. Add more terms in the keywords to better summarize the scope of the paper.
Comment 3. Carefully check the journal’s template, particularly removing the grid of figures.
Comment 4. Section 1 Introduction:
(a) Why was Figure 1 necessary as the background information?
(b) A literature review is missing.
(c) Clearly provide the research contributions in point form.
Comment 5. Section 2 Materials and Methods:
(a) Enhance the organization of “Computational Simulations”, “Geometry Creation”, etc. using subsections 2.1, 2.2, …
(b) If screenshots are used, ensure high resolutions of figures. The authors should enlarge the file to 200-300% to ensure no content is blurred. Otherwise, the content cannot be visualized and reviewed.
(c) Too many citations were used in one sentence without further elaboration “following the same procedure used 88 in our previous modeling [8-12]”
(d) Table 1, the title is about “… in each model”; however, the first column is named “Part”.
(e) Based on the content in this section, the details of the finite element modeling are insufficient.
Comment 6. Section 3 Results:
(a) The format of subfigures is not appropriate in Figures 7-9. Please refer to the journal’s template.
(b) More results should be shared to analyze the performance of the model.
Comment 7. Section 4 Discussion: What is the aim of this section compared with Section 3?
Comment 8. Compare the performance between your work and the existing works.

Comments on the Quality of English Language

There are some typos and improper use of English.

Author Response

(The authors gave the same response as above.)

Reviewer 3 Report

Comments and Suggestions for Authors

The manuscript entitled Advancements In Finite Element Modeling for Cardiac Device Leads And 3D Heart Models is an original article. The authors analyzed the advancements in finite element modeling for cardiac leads and 3D heart models, leveraging computational simulations to assess lead behavior over time. They concluded that there is a close agreement between experimental and simulated data, highlighting the effectiveness of simulation in predicting lead performance.

The article is important because in interventional cardiology (and not only), simulation is very important.  

The article is well written. The materials and methods are well described by the authors. The results are presented in detail. Discussions seems too long.

I have two minor issues.

What are the study limitations?

What do you mean by ASTM, in abstracts? Please explain this acronym.

Author Response

(The authors gave the same response as above.)

Reviewer 4 Report

Comments and Suggestions for Authors

The scope of work is relevant, interesting and has a real application area. Of course, work in the biomedical area requires prior analyses and these analyses are the subject of a peer-reviewed paper. The aim of the study is clear and concerns a critical area of the implementation of the artificial heart - the interface between the artificial heart and the human body. Bravo!

A review of the literature is sufficient, although for such a task it would be useful to have a more extensive excerpt on the analysis of the "mechanical" properties of the tissues of the heart and surrounding the heart (3 positions is a bit low).

On the one hand, the scarcity of data on tissues limits analyses and, on the other hand, indicates the necessary supplementary work that should be carried out in the biomedical and technical areas. I know, such works are difficult to carry out - and that is why they are very important.

The proposed model and software for the implementation of the calculations are correct. The method of introducing boundary conditions is interesting.

Unfortunately, as a reviewer, I have something to complain about. And so: in graphs 7-10, 12,14, 17, the description of what is on the axis requires intensive zooming. Something needs to be done about it. Probably make them look as figure 6. Figure 16 does not have an proper axis descriptions.

Besides, it was a pleasure to read the submitted work.

Author Response

(The authors gave the same response as above.)

Round 2

Reviewer 1 Report

Comments and Suggestions for Authors

Accept

Author Response

Dear Reviewer,

Thank you for your insightful comments and positive feedback on our article. We appreciate your efforts in reviewing our work and are grateful for your constructive suggestions.

Reviewer 2 Report

Comments and Suggestions for Authors

Some comments remain unaddressed. Please confirm the following issues:
Follow-up Comment 1 (a): Define the acronym ASTM. Changes should be applied in the revised paper.
Follow-up Comment 5 (b): If screenshots are used, ensure high resolutions of figures. The authors should enlarge the PDF file to confirm no content is blurred. Most of the content cannot be visualised in many figures.

There are some new comments on the updated content:
New Comment 1: The in-text citations must be ordered in ascending order.
New Comment 2: Regarding the newly added literature review, rewrite the paragraphs because they are not focused on recently published articles. 

Author Response

(The authors gave the same response as above.)
